# Intermittent preventive treatment with sulphadoxine-pyrimethamine but not dihydroartemisinin-piperaquine modulates the relationship between inflammatory markers and adverse pregnancy outcomes in Malawi

Kaylene Cheng[1☉], Elizabeth H. Aitken [2,3☉*], Wina Hasang[2], Niamh Meagher [2], David J. Price [2,4], Mwayiwawo Madanitsa [5,6], Victor Mwapasa[7], Kamija S. Phiri [7,8], James Dodd [6], Feiko O. ter Kuile[6], Stephen J. Rogerson[1,2]

1 Department of Medicine (RMH), The Peter Doherty Institute of Infection and Immunity, University of Melbourne, Melbourne, Victoria, Australia, 2 Department of Infectious Diseases, The Peter Doherty Institute of Infection and Immunity, University of Melbourne, Melbourne, Victoria, Australia, 3 Department of Microbiology and Immunology, The Peter Doherty Institute of Infection and Immunity, University of Melbourne, Melbourne, Victoria, Australia, 4 Centre for Epidemiology & Biostatistics, Melbourne School of Population & Global Health, University of Melbourne, Melbourne, Victoria, Australia, 5 Department of Clinical Sciences, Academy of Medical Sciences, Malawi University of Science and Technology, Thyolo, Malawi, 6 Department of Clinical Sciences, Liverpool School of Tropical Medicine, Liverpool, United Kingdom, 7 Department of Epidemiology and Biostatistics, School of Global and Public Health, Kamuzu University of Health Sciences, Blantyre, Malawi, 8 Training and Research Unit of Excellence, Blantyre, Malawi

☉ These authors contributed equally to this work.
* elizabeth.aitken@unimelb.edu.au

## Abstract

Women in malaria-endemic areas receive sulphadoxine-pyrimethamine (SP) as Intermittent Preventive Treatment in Pregnancy (IPTp) to reduce malaria. While dihydroartemisinin-piperaquine (DP) has superior antimalarial properties as IPTp, SP is associated with superior fetal growth. As maternal inflammation influences fetal growth, we investigated whether SP alters the relationship between inflammation and birth outcomes. We measured C-reactive protein (CRP) and alpha-1-acid glycoprotein (AGP) at enrollment (16–28 gestation weeks (gw)), visit 3 (24–36 gw) and delivery in 1319 Malawian women randomized to receive monthly SP, DP, or DP and single-dose azithromycin (AZ) in the IMPROVE trial (NCT03208179). Logistic regression was used to assess the relationship between adverse outcomes, inflammation, and treatment arm. Elevated AGP at enrollment was associated with adverse birth outcome (aRR 1.40, 95% CI: 1.15, 1.70), with similar associations observed across treatment arms, exceptions being that elevated AGP was associated with low maternal weight gain in SP recipients (aRR 1.94, 95% CI: 1.36, 2.76) and with small for gestational age in DP+AZ recepients (aRR 1.49, 95% CI 1.02, 2.17). At visit 3 there were few associations between inflammation andoutcomes. At delivery, women with elevated AGP receiving either DP or DP+AZ had an increased risk of adverse birth outcomes (aRR 1.60, 95% CI: 1.28, 2.00), including low birth weight, pre-term birth and foetal loss, this was

**Data Availability Statement:** CRP and AGP levels are available in the Supporting Information files. In addition, CRP and AGP levels along with clinical data that support the findings of this study have been uploaded to the WorldWide Antimalarial Research Network (WWARN) in the Infectious Diseases Data Observatory (IDDO). Data can be requested from WWARN (https://www.iddo.org/wwarn/accessing-data), IDDO Data Contribution ID:WFETV.

**Funding:** This work was supported, in whole or in part, by the Bill & Melinda Gates Foundation (INV-002781 to SJR and FOtK). Under the grant conditions of the Foundation, a Creative Commons Attribution 4.0 Generic License has already been assigned to the Author Accepted Manuscript version that might arise from this submission. The IMPROVE trial was funded through the European and Developing Countries Clinical Trials Partnership (EDCTP-2) programme, with assistance from the Swedish International Development Cooperation Agency, and by the United Kingdom Joint Global Health Trials (JGHT) scheme, funded by as well as by funding National Institute of Health Research (NIHR), the U.K. Foreign Commonwealth and Development Office (FCDO), the U.K. Medical Research Council (MRC), and Wellcome (TRIA2015-1092 to FOtK). The funders had no role in study design, data collection and analysis, decision to publish or preparation of the manuscript.

**Competing interests:** The authors have declared that no competing interests exist.

not seen in women receiving SP (aRR 0.82, 95% CI: 0.54, 1.26). The risk of an association between elevated AGP and adverse birth outcome was higher in those receiving DP or DP +AZ compared to those receiving SP (aRR 1.95, 95% CI: 1.21, 3.13). No clear associations between CRP and adverse outcomes were observed. AGP identified women at risk of adverse pregnancy outcomes. SP modifies the relationship between inflammatory biomarkers and adverse outcomes. Our findings provide insights into potential mechanisms by which SP may improve pregnancy outcomes.

## Introduction

In low- and middle-income countries (LMIC), adverse birth outcomes are prevalent, including fetal loss due to miscarriage and stillbirth, preterm birth (PTB; <37 weeks' gestation), low birth weight (LBW; <2500g) and small for gestational age (SGA; an infant born with a birth weight below the 10th centile of the intergrowth-21st standards) [1–3]. In 2010, around 32.4 million infants from LMICs were born SGA [4], and of 18 million infants born with LBW, 41% were PTB [4].

Pregnant women have increased susceptibility to malaria infections. In 2019 in Africa, malaria in pregnancy resulted in an estimated 900,000 LBW deliveries due to a mix of SGA and PTB [5, 6]. In LMICs, sexually transmitted infections, HIV, and reproductive tract infections also contribute to adverse pregnancy outcomes [7–9].

To decrease the burden of malaria in pregnancy, the World Health Organization (WHO) recommends intermittent preventive treatment in pregnancy (IPTp) in high malaria transmission areas of Africa [10]. Unfortunately, high-grade resistance to sulphadoxine-pyrimethamine (SP), the only antimalarial currently recommended by WHO for IPTp, is now widespread [11]. Potential alternatives include dihydroartemisinin-piperaquine (DP) and DP plus azithromycin (AZ). A mediation analysis suggested that, although DP was a better antimalarial than SP, the latter had a greater positive effect on birth weight, probably due to the non-malarial effects of SP on fetal growth [12]; the underlying mechanisms are largely unknown.

Markers of inflammation in pregnancy, such as circulating concentrations of C-reactive protein (CRP), have been associated with placental malarial infections and correlated with placental parasite density [13]. CRP has also been associated with fetal growth restriction and PTB [14–16]. Increased concentrations of alpha-1 acid glycoprotein (AGP), another biomarker of inflammation, have been associated with adverse pregnancy outcomes in malaria-endemic areas, including PTB, foetal growth restriction and LBW [17, 18].

Understanding the pathways associated with adverse outcomes is crucial to find ways to prevent them. Our previous studies in pregnant women from Papua New Guinea (PNG) showed that the relationship between inflammation and fetal outcome is moderated by antimalarial treatment [17]. This study showed strong associations between inflammation and adverse outcomes in participants receiving a single dose of SP and chloroquine but not in those receiving multiple doses of SP+AZ. Therefore, it was hypothesised that SP and AZ might dampen the relationship between inflammatory biomarkers and adverse birth outcomes. IMPROVE was a randomized three arm (SP, DP or DP+AZ) trial of IPTp in areas with SP resistance in Malawi, Kenya and Tanzania which showed that IPTp treatment with DP or DP +AZ was associated with higher risk of adverse birth or pregnancy outcomes compared to IPTp with SP [19]. In this study, we investigated whether AGP and CRP are markers of adverse birth outcomes in a subset of the IMPROVE cohort, pregnant Malawian women. We further

asked whether IPTp with SP, rather than DP or DP+AZ, may alter the relationship between inflammation and adverse birth outcomes, to investigate potential mechanisms for the malaria-independent effects of IPTp with SP on fetal growth [12].

## Methods

### Ethics statement

This work was approved by the Melbourne Health Human Research Ethics Committee (HREC/56534/MH-2020) and the College of Medicine Research and Ethics Committee (COMREC 01/19/2578). Written informed consent was obtained from participants.

### Study design

This study was nested within a larger IPTp trial in Malawi (IMPROVE, NCT03208179) [19]. From 16–28 weeks' gestation, participants were randomised to receive monthly IPTp with SP (three tablets of SP, total 1500 mg of sulphadoxine and 75 mg of pyrimethamine), DP+AZ (monthly DP, three to five tablets of 40 mg of dihydroartemisinin and 320 mg of piperaquine based on bodyweight daily for three days and a single two-day course of AZ 1 g daily), or DP (monthly DP as above, plus two days of AZ placebo at enrolment). Randomisation and follow up procedures are described elsewhere [19].

Women were screened for inclusion in the IPTp trial between March 29, 2018 and July 15, 2019. At enrolment, maternal age, gravidity, study site, and measures of socioeconomic status (calculated using principal component analysis) [20] and bed net use were documented, and maternal height and mid-upper arm circumference were measured. Maternal weight was measured at each visit, and average weekly gestational weight gain was calculated. Gestational age was determined by ultrasound. Plasma from venous blood samples collected at enrolment (week 16–28), visit 3 (week 24–36) and delivery was frozen at -80˚C and sent to The Doherty Institute, Melbourne, Australia, for analysis of inflammatory markers. Plasma samples were accessed on 29 July, 2021 and corresponding clinical data was accessed on 21 September, 2021. All available plasma samples for the three selected timepoints from Malawi were used.

*Plasmodium spp*. infection was determined by microscopy of Giemsa-stained blood smears and polymerase chain reaction (PCR) of peripheral blood samples. Haemoglobin concentration in peripheral blood was measured at enrolment and delivery by HemoCue (HemoCue AB, Ängelholm, Sweden). Gestational age, birth weight, birth outcome and sex were recorded at birth.

### Outcome measures

The primary outcome for this study was a composite measure of adverse birth outcome, defined as LBW, PTB, SGA, foetal loss (miscarriage or stillbirth) or neonatal death (within 28 days of birth) [19]. Secondary outcomes included the relationship between inflammation and each individual adverse birth outcome included in the composite primary outcome, and the association between inflammatory markers and low maternal weight gain (defined as <0.164 Kg/week, representing the lowest quartile of the IMPROVE study population) [17]. The relationship between inflammation and continuous outcome variables growth z-score, gestational age at birth, birth weight and maternal weight gain was also investigated [19].

### Measuring CRP and AGP

Human CRP and AGP concentrations in plasma were measured using DuoSet ELISA kits from R&D systems (DY1707 and DY3694). To facilitate the measurement of the high number

of samples the protocols were modified for use with 384 microwell plates (Thermo Scientific NUNC 384).

To measure CRP, wells were coated overnight with 40 μL of Mouse Anti-Human CRP capture antibody in phosphate buffer saline (PBS) at 1 μg/mL. After washing thrice with 100 μL of wash buffer (0.05% Tween 20 from Sigma-Aldrich in PBS) using a Thermo Scientific Multidrop Combi Reagent Dispenser, wells were blocked with 100 μL of reagent diluent (1% bovine serum albumin (BSA) diluted in PBS) at room temperature (RT) for one hour and washed again. Reagent diluent was used to dilute plasma samples, 40 μL pre-diluted to 1:10,000, a Liquicheck Immunology control (Biorad, 1:80,000; to monitor plate to plate variation) and recombinant human CRP standard (doubling dilutions from 1000 pg/mL). These were incubated at RT for two hours. After washing, 40 μL biotinylated mouse anti-human CRP detection antibody (1:360 in reagent diluent) was added and incubated at RT for two hours. Following further washing, 40 μL streptavidin-horse radish peroxidase (HRP) diluted to 1:200 was added to each well and left at RT in the dark for 20 minutes. Next, the plates were washed, and 40 μL 3,3',5,5'-Tetramethylbenzidine (TMB) substrate (BD OptEIA) was added to each well and plates were incubated in a FLUOstar Omega plate reader. Forty μL of 2N $H_2SO_4$ was added to each well to stop the reaction when an optical density value of 0.9 at 650 nm was obtained. Plates were then read again at 450 nm. CRP levels in samples and the Liquicheck control were extrapolated from the standard curve using Graphpad PRISM 9 software. Samples above the standard curve were rerun at dilutions between 1:20,000 to 1:600,000. Any plate where the Liquicheck Immunology control was out by >25% was also rerun.

A similar protocol was followed for AGP. Mouse Anti-Human AGP capture antibody and Biotinylated Goat Anti-Human AGP detection antibody from the AGP DuoSet ELISA kits from R&D systems were diluted to 1:360 per well. Plasma samples were diluted to 1:1,000,000 in reagent diluent the day before the experiment. The Liquicheck Immunology control (to monitor plate-to-plate variation) was diluted 1:2,000,000. Samples above the standard curve were rerun at 1:2,000,000 dilution. A solution containing a known concentration of AGP from human plasma (Sigma-Aldrich) was used as an additional standard. Any plate was rerun if the Liquicheck Immunology control was out by >20%.

## Statistical analysis

Maternal characteristics at enrolment and delivery, social or behavioural factors, and birth outcomes were reported for the overall cohort and by treatment arm. Median CRP and AGP levels at each time point were plotted and compared using Wilcoxon matched-pairs signed rank test. Median CRP and AGP in the three treatment arms were compared at each timepont using Kruskal-Wallis test. Changes in inflammatory markers in individual women were calculated by subtracting levels at enrollment from those at visit 3 and delivery, these changes in inflammatory markers in the different treatment arms were then compared using a t-test.

Elevated CRP and AGP were defined as ≥5 mg/L and the top quartile of measured levels (≥0.345 g/L) [15, 17], respectively. To evaluate the association between elevated CRP or AGP and adverse outcomes, the Stata adjrr package [21] was used to estimate adjusted risk ratios (aRR) and the associated 95% confidence intervals (CI) and p-value from adjusted logistic regression models. Adjusted linear regression was used to evaluate the association between inflammatory markers and continuous birth and maternal outcomes, and the estimated marginal effect of elevated CRP or AGP, in addition to the associated 95% CI and p-value, was reported for each treatment arm. All models adjusted for maternal age, maternal height, mid-upper arm circumference, gravidity, number of antenatal visits and study site, and included an interaction term between inflammatory marker and treatment arm, and participants were

only included in models if they had no missing data for the variables of interest. To explore possible differences in risks of an association between inflammation and outcome in the different treatment arms the ratio of risk ratios was calculated. No adjustments for multiple testing were done when calculating P-values. P-values ≤0.05 were defined as significant and highlighted in bold text. Data were analyzed using Stata 17 [22].

## Results

### Study population and pregnancy outcomes

Of 1404 study participants in Malawi, outcome and CRP and AGP data were available at one or more time points for 1319 women. At enrolment, the mean gestational age was 149 days, 29% were in their first pregnancy, 40% were anaemic (haemoglobin <11.0 g/dL), and 19% were infected with *Plasmodium spp.* by PCR (Table 1). At delivery, participants had attended a mean of 4.5 antenatal care visits. Adverse outcomes were frequent with 28.3% of births having SGA, LBW, PTB, foetal loss and/or neonatal death, while low gestational weight gain affected 27.5% of participants (Table 1).

**Table 1. Participant characteristics and birth outcomes.**

| | | SP | DP | DP+AZ | Overall |
|---|---|---|---|---|---|
| | | N = 439 | N = 442 | N = 438 | N = 1319 |
| **Maternal characteristics at enrolment (week 16–28)** | | | | | |
| Age[1] (years) | | 24.34 (5.82) | 24.76 (5.94) | 24.30 (5.58) | 24.43 (5.78) |
| Height[2] (cm) | | 157.11 (6.07) | 157.57 (6.62) | 157.75 (7.37) | 157.48 (6.71) |
| Mid-upper arm circumference[3] (cm) | | 26.16 (3.12) | 26.33 (3.04) | 26.27 (2.92) | 26.26 (3.03) |
| Gravidity[4] | Primigravid | 141 (32) | 125 (28) | 111 (25) | 377 (29) |
| | Secundigravid | 106 (24) | 120 (27) | 130 (30) | 356 (27) |
| | Multigravid | 192 (44) | 195 (44) | 195 (45) | 582 (44) |
| Gestational age, days | | 148 (21) | 149 (22) | 150 (22) | 149 (22) |
| Haemoglobin levels[5] (g/dL) | | 11.14 (1.48) | 11.27 (1.36) | 11.22 (1.44) | 11.21 (1.43) |
| Anaemia[5] | | 178 (41.2) | 168 (38.4) | 175 (39.9) | 520 (39.9) |
| Peripheral parasitaemia | PCR[6] | 76 (18.9) | 73 (17.9) | 82 (20.3) | 231 (19.0) |
| | Microscopy[7] | 51 (11.7) | 49 (11.1) | 66 (14.1) | 166 (12.6) |
| Bed net use[8] | | 282 (64.2) | 291 (66.0) | 283 (64.6) | 856 (65.0) |
| Socioeconomic status | Low | 311 (70.8) | 306 (69.2) | 310 (70.8) | 927 (70.3) |
| | Medium | 72 (16.4) | 81 (18.3) | 78 (17.8) | 231 (17.5) |
| | High | 56 (12.8) | 55 (12.4) | 50 (11.4) | 161 (12.2) |
| **Maternal characteristics at visit 3** | | | | | |
| Peripheral parasitemia | PCR[9] | 7 (5.79) | 3 (2.38) | 7 (5.51) | 17 (4.55) |
| | Microscopy[10] | 19 (4.52) | 7 (1.64) | 2 (0.47) | 28 (2.21) |
| **Maternal characteristics at delivery** | | | | | |
| Haemoglobin levels[11] (g/dL) | | 11.74 (1.61) | 11.77 (1.64) | 11.75 (1.49) | 11.76 (1.58) |
| Anaemia[11] | | 108 (27) | 123 (30) | 104 (26) | 335 (28) |
| Gestational weight gain[12] (average kg/week) | | 0.297 (0.224) | 0.266 (0.227) | 0.265 (0.232) | 0.276 (0.228) |
| Low gestational weight gain[12] | | 98 (23.0) | 130 (30.1) | 126 (29.4) | 354 (27.5) |
| Total ANC visits | | 4.5 (1.0) | 4.5 (1.1) | 4.5 (1.0) | 4.5 (1.0) |
| Placental parasitemia at delivery (Microscopy)[13] | | 1 (0.26) | 4 (1.02) | 5 (1.28) | 10 (0.85) |
| Peripheral parasitaemia at delivery | PCR[14] | 32 (8.7) | 20 (5.5) | 28 (7.8) | 80 (7.3) |
| | Microscopy[15] | 16 (3.9) | 8 (1.9) | 12 (3.0) | 36 (2.9) |
| **Birth outcomes** | | | | | |
| Birth weight[16] (kg) | | 3.04 (0.48) | 2.97 (0.50) | 2.97 (0.46) | 2.99 (0.48) |
| Gestational age at birth[17] (days) | | 273.7 (16.6) | 273.5 (15.8) | 273.5 (15.7) | 273.6 (16.0) |

*(Continued)*

**Table 1.** (Continued)

| | | | | |
|---|---|---|---|---|
| Weight-Z score[18] | -0.35 (1.06) | -0.52 (1.07) | -0.50 (1.04) | -0.46 (1.06) |
| Any adverse birth outcome[19] | 107 (25.3) | 129 (30.1) | 125 (29.4) | 361 (28.3) |
| Low birth weight[16] | 32 (7.8) | 55 (13.4) | 47 (11.5) | 134 (10.9) |
| Small for gestational age[20] | 67 (16.3) | 90 (21.8) | 95 (23.2) | 252 (20.4) |
| Pre-term birth[21] | 38 (9.2) | 32 (7.8) | 32 (7.8) | 102 (8.3) |
| Foetal loss[22] | 8 (1.9) | 9 (2.1) | 8 (1.9) | 25 (2.0) |
| Neonatal death[23] | 4 (1.0) | 5 (1.3) | 2 (0.5) | 11 (0.9) |

Data are mean and standard deviation (SD) for continuous variables or numerator/denominator (%) for categorical/binary variables. Intermittent prevention treatment in pregnancy; DP = Dihydroartemisinin-piperaquine; SP = Sulphadoxine-pyrimethamine; AZ = Azithromycin; PCR = Polymerase chain reaction; ANC ante-natal care. Socioeconomic status defined in terciles low, medium or high based on socioeconomic terciles calculated using principle component analysis on data from all Malawi, Kenya and Tanzania [20]. Low birth weight defined as <2500g; Preterm birth defined as <37 weeks' gestation; Foetal loss is defined as miscarriage or stillbirth; Small for gestational age defined as birth weight lower than the 10th centile of the intergrowth-21st standards [1]. Low gestational weight gain was defined as the bottom quartile of the IMPROVE study population (<164 g/week). Anaemia defined as haemoglobin concentration <11 g/dL. 1, Missing 8 women age; 2,Missing 7 women height; 3 Missing 1 woman mid-upper arm circumference; 4, Missing 4 gravidity; 5, Missing 14 Hb enrollment & anaemia; 6, Missing 5 PCR enrolment; 7, Missing 4 microscopy enrolment; 8, 1 missing data bednet; 9, 945 missing PCR visit 3; 10, 51 missing microscopy visit 3; 11, 121 missing Hb delivery & anaemia data; 12, 32 missing gestational weight gain/week & low gestational weight gain data; 13, 146 missing placental microscopy at delivery; 14, 229 missing PCR data at delivery; 15, 89 missing microscopy data at delivery; 16, 87 missing birth weight data at delivery; 17, 53 missing gestation length data; 18, 103 missing z-scores; 19, 42 missing data for any adverse birth outcome; 20, 85 missing small for gestational age; 21, 43 missing pre-term birth data;; 22, 42 missing data for foetal loss; 23, 133 missing data for neonatal loss.

## CRP and AGP concentrations during pregnancy

CRP concentrations were elevated in 35.7% (475/1332), 16.4% (181/1102) and 74.8% (818/1094) of women at enrolment, visit 3 and delivery, respectively. Regarding AGP, 31.3% (417/1332), 9.4% (104/1102) and 32.5% (356/1094) of women had AGP concentrations in the upper quartile at enrolment, visit 3 and delivery, respectively (Table 2). Concentrations of AGP dropped between enrolment (median 0.27 g/L, interquartile range [IQR]: 0.19–0.39) and week 24–32 (median 0.18 g/L, IQR: 0.13–0.25) and then rose again at delivery (median 0.28 g/L, IQR: 0.20–0.39) (Fig 1A). CRP concentrations also dropped between enrolment (median 3.37 mg/L, IQR: 1.79–6.77) and week 24–32 (median 1.70 mg/L, IQR: 0.79–3.59) but then rose 5-fold at delivery (median 9.28 mg/L, IQR: 5.0–19.28) (Fig 1B). Between enrolment and week 24–32, levels of both AGP (Fig 1C) and CRP (Fig 1D) fell most in women receiving DP+AZ,

**Table 2. Women with elevated CRP and/or AGP levels during pregnancy, N = 1332.**

| | | SP | | DP | | DP+AZ | | Overall | |
|---|---|---|---|---|---|---|---|---|---|
| | | n | (%) | n | (%) | n | (%) | n | (%) |
| Enrolment (Week 16–28) | Elevated CRP | 155 | (34.9) | 163 | (36.7) | 157 | (35.4) | 475 | (35.7) |
| | Elevated AGP | 130 | (29.3) | 148 | (33.3) | 139 | (31.3) | 417 | (31.3) |
| Visit 3 | Elevated CRP | 70 | (19.3) | 59 | (15.9) | 52 | (14.2) | 181 | (16.4) |
| | Elevated AGP | 38 | (10.5) | 34 | (9.1) | 32 | (8.72) | 104 | (9.4) |
| Delivery | Elevated CRP | 273 | (75.0) | 281 | (76.2) | 264 | (73.1) | 818 | (74.8) |
| | Elevated AGP | 110 | (30.2) | 124 | (33.6) | 122 | (33.8) | 536 | (32.5) |

Elevated CRP was defined as ≥5 mg/L and elevated AGP was defined as ≥ the top quartile of all measured levels (≥ 0.345 g/L) [15, 17]. Visit 3, third treatment with IPTp 24–36 gestation weeks (gw). 1332 women had AGP and/or CRP measured, however only 1319 of these had relevant clinical data to be included in the final analysis.

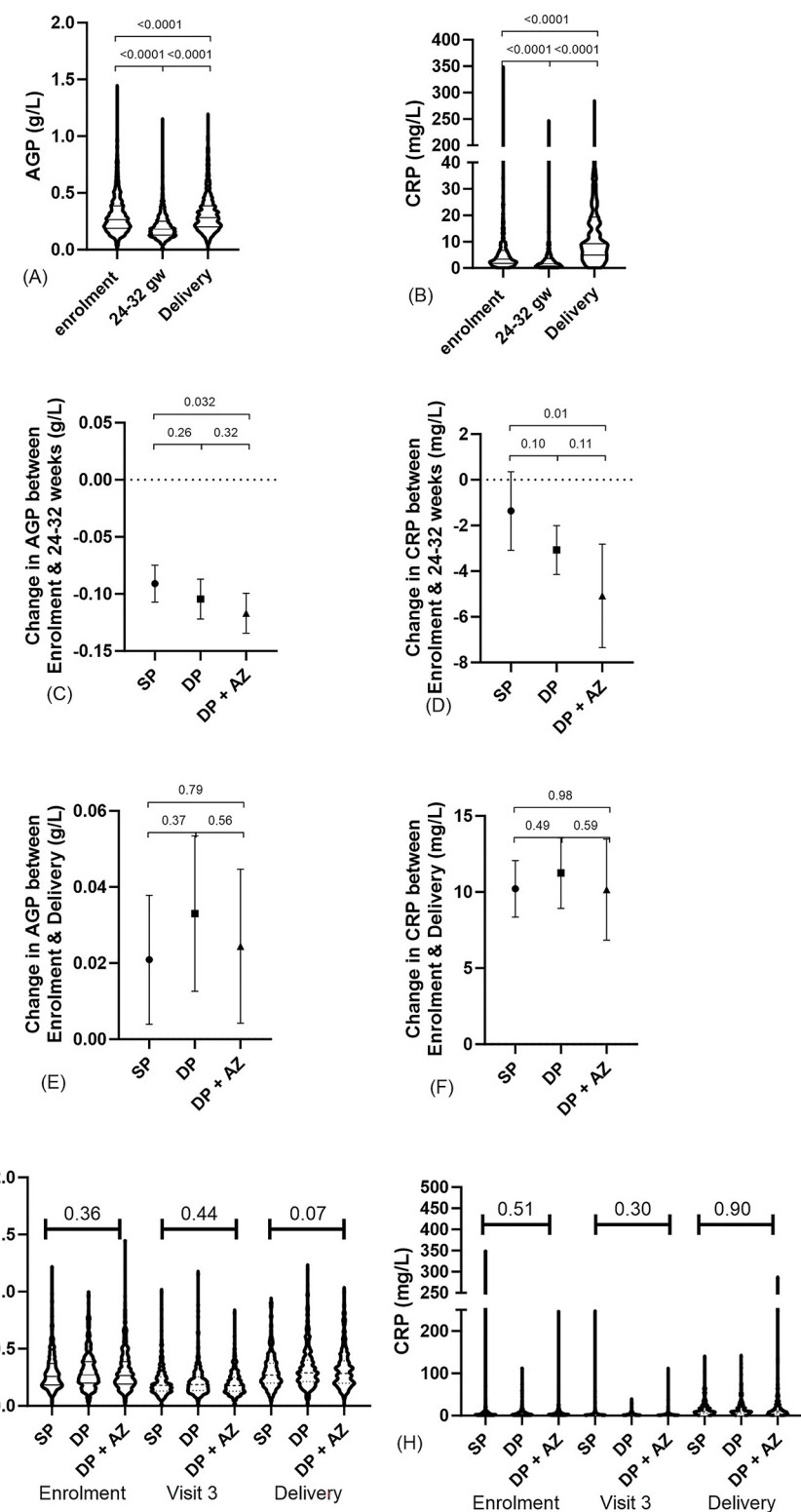

**Fig 1.** Distribution of AGP and CRP (A,B, G &H) and changes in AGP and CRP between enrolment and 24–32 weeks (C&D) and enrolment and delivery (E&F) by treatment arm. Wilcoxon matched-pairs signed rank test was used to assess differences between time points. Unpaired t-test was used to assess differences in changes in inflammatory markers between treatment arms. Kruskall-wallis was used to assess whether levels of inflammatory markers were similar in all treatment arms at each time point. Lines represent median and IQR (A, B, G & H) or Mean, 95% CI of changes between time points (C—F).

**Table 3. Elevated CRP and AGP levels at enrolment and risks of individual adverse outcomes.**

| Enrolment (Week 16–28) | | LBW | | SGA | | PTB | | Foetal loss | | Neonatal death | | Any adverse foetal outcome | | Low Gestational Weight Gain | |
|---|---|---|---|---|---|---|---|---|---|---|---|---|---|---|---|
| | | aRR (95% CI) | P | aRR (95% CI) | P | aRR (95% CI) | P | aRR (95% CI) | P | aRR (95% CI) | P | aRR (95% CI) | P | aRR (95% CI) | P |
| | CRP | 1.27 (0.92, 1.76) | 0.15 | 1.15 (0.91, 1.45) | 0.25 | 1.22 (0.86, 1.74) | 0.27 | 1.18 (0.53, 2.62) | 0.69 | ** | | **1.21 (1.01, 1.45)** | **0.04** | 0.97 (0.80, 1.17) | 0.75 |
| | AGP | 1.36 (0.96, 1.93) | 0.09 | **1.34 (1.04, 1.72)** | **0.02** | 1.23 (0.83, 1.82) | 0.31 | 1.03 (0.42, 2.55) | 0.95 | ** | | **1.40 (1.15, 1.70)** | **0.001** | **1.23 (1.01, 1.51)** | **0.04** |

The composite adverse birth outcome variable includes: Low birth weight (LBW) defined as < 2500 g, preterm birth (PTB), defined as <37 gestational weeks, foetal loss defined as a non-live birth, small for gestational age (SGA) defined as birth weight lower than the 10th centile of the intergrowth-21st standards [1] and/or neonatal loss. Low gestational weight gain was defined as the bottom quartile of the IMPROVE study population (<164 g/week). Adjusted Risk Ratios (aRR) and 95% confidence intervals were estimated using logistic regression and adjusted for maternal age, maternal height, mid-upper arm circumference, gravidity, number of antenatal visits, study site, and included an interaction term between inflammatory marker and treatment arm. Significance is defined as P <0.05 and significant findings are highlighted with bold text. ** aRRs were not always able to be estimated for neonatal death as this outcome occurred infrequently.

and least in women receiving SP. Changes in inflammatory markers between enrolment and delivery were similar in all three arms (Fig 1E & 1F). Levels of CRP and AGP and proportions of women with elevated AGP or CRP were similar in each of the three treatment arms at all three timepoints (Fig 1 & Table 2).

## Associations between inflammation at baseline and adverse outcomes

The aRR for those with elevated AGP or CRP at enrolment for each individual adverse outcome is presented in Table 3. Elevated AGP was associated with an increased risk of an adverse birth outcome (aRR 1.40, 95% CI: 1.15, 1.70) and was specifically associated with an increased risk of SGA (aRR 1.34, 95% CI: 1.04, 1.72). Elevated AGP was also associated with a 23% increased risk of low maternal weight gain (aRR 1.23, 95% CI: 1.01, 1.51) (Table 3). Elevated CRP was associated with an increased risk of adverse birth outcomes (aRR 1.21, 95% CI: 1.01, 1.45), but was not significantly associated with any individual adverse outcome.

## Associations between inflammation and adverse outcomes in the different treatment arms

At enrolment in women receiving DP+AZ, elevated AGP was associated with increased risk of an adverse outcome (aRR 1.49, 95% CI: 1.09, 2.03), and in this arm elevated AGP at enrolment was specifically associated with increased risk of SGA (aRR 1.49, 95% CI: 1.02, 2.17) (Table 4). At Visit 3, elevated CRP was associated with adverse outcomes in the SP arm (aRR 1.50, 95% CI: 1.00, 2.23), with a similar association seen in the DP arm (aRR 1.43, 95% CI: 0.99, 2.08). At delivery, elevated AGP was strongly associated with an increased risk of adverse birth outcome in women who received either DP or DP+AZ (aRR 1.60, 95% CI: 1.28, 2.00) but not in the SP arm (aRR 0.82, 95% CI: 0.54, 1. 26) (Table 4).

Linear regression analysis showed that among women receiving DP+AZ, those with elevated AGP at enrolment had babies weighing on average 130 g less (95% CI: -230, -3) and born on average 3.44 days earlier (95% CI: -6.70, -0.17) (Table 5) compared to women without elevated AGP. In women receiving SP, elevated AGP at enrollment was associated with 170 g lower birth weight on average (95% CI: -270, -7), and babies born an average of 3.51 days earlier (95% CI: -6.87, -0.15) suggesting similar associations between inflammation at enrolment and birth outcomes in the SP and DP+AZ arms (Table 5). There was no association with AGP

**Table 4. Elevated CRP and AGP levels during pregnancy and risks of individual adverse outcomes, by treatment arm.**

| SP | | LBW | | SGA | | PTB | | Foetal loss | | Neonatal loss | | Any adverse foetal outcome | | Low maternal weight gain | |
|---|---|---|---|---|---|---|---|---|---|---|---|---|---|---|---|
| | | aRR (95% CI) | P | aRR (95% CI) | P | aRR (95% CI) | P | aRR (95% CI) | P | aRR (95% CI) | P | aRR (95% CI) | P | aRR (95% CI) | P |
| **Enrolment (Week 16–28)** | CRP | 1.30 (0.65, 2.62) | 0.45 | 1.25 (0.78, 1.99) | 0.35 | 1.26 (0.71, 2.22) | 0.43 | 0.40 (0.05, 3.03) | 0.37 | 1.21 (0.16, 9.03) | 0.86 | 1.23 (0.88, 1.72) | 0.23 | 0.86 (0.59, 1.25) | 0.43 |
| | AGP | 1.22 (0.58, 2.54) | 0.60 | 1.30 (0.80, 2.12) | 0.29 | 1.10 (0.60, 2.03) | 0.76 | 1.41 (0.33, 5.95) | 0.63 | 10.00 (0.96, 104.90) | 0.055 | 1.40 (0.99, 1.98) | 0.058 | **1.94 (1.36, 2.76)** | **0.0002** |
| **Visit 3** | CRP | 2.06 (0.97, 4.38) | 0.06 | **1.80 (1.08, 2.97)** | **0.023** | 1.08 (0.50, 2.33) | 0.85 | ** | | 2.00 (0.027, 14.73) | 0.49 | **1.50 (1.00, 2.23)** | **0.049** | 1.22 (0.78, 1.92) | 0.39 |
| | AGP | **2.47 (1.05, 5.82)** | **0.039** | 1.37 (0.68, 2.79) | 0.38 | 1.28 (0.50, 3.32) | 0.61 | 6.57 (0.57, 76.2) | 0.13 | 3.51 (0.43, 28.9) | 0.24 | 1.48 (0.98, 2.44) | 0.13 | 1.43 (0.82, 2.47) | 0.21 |
| **Delivery** | CRP | 1.19 (0.51, 2.78) | 0.69 | 0.76 (0.45, 1.26) | 0.28 | 0.87 (0.43, 1.75) | 0.70 | 1.17 (0.12, 11.48) | 0.90 | 0.27 (0.02, 4.01) | 0.34 | 0.85 (0.57, 1.27) | 0.44 | 1.14 (0.72, 1.80) | 0.58 |
| | AGP | 0.87 (0.40, 1.91) | 0.73 | 0.83 (0.48, 1.43) | 0.50 | 0.94 (0.46, 1.92) | 0.87 | 6.67 (0.72, 62.06) | 0.10 | ** | | 0.82 (0.54, 1.26) | 0.37 | 1.42 (0.97, 2.08) | 0.074 |
| **DP** | | | | | | | | | | | | | | | |
| **Enrolment (Week 16–28)** | CRP | 1.15 (0.69, 1.91) | 0.59 | 0.98 (0.66, 1.45) | 0.91 | 1.12 (0.60, 2.10) | 0.72 | 1.93 (0.53, 7.00) | 0.32 | 0.40 (0.04, 3.70) | 0.42 | 1.08 (0.80, 1.45) | 0.63 | 1.03 (0.76, 1.40) | 0.83 |
| | AGP | 1.43 (0.86, 2.39) | 0.67 | 1.22 (0.82, 1.81) | 0.33 | 1.25 (0.66, 2.37) | 0.49 | 1.17 (0.30, 4.62) | 0.82 | 2.19 (0.36, 13.55) | 0.40 | 1.31 (0.97, 1.77) | 0.078 | 1.16 (0.85, 1.58) | 0.35 |
| **Visit 3** | CRP | **2.44 (1.36, 4.37)** | **0.0028** | 1.43 (0.90, 2.28) | 0.13 | 1.35 (0.55, 3.37) | 0.51 | ** | | 1.72 (0.016, 18.7) | 0.65 | 1.43 (0.99, 2.08) | 0.059 | 1.09 (0.73, 1.63) | 0.66 |
| | AGP | 1.52 (0.67, 3.45) | 0.32 | 1.11 (0.57, 2.16) | 0.76 | 1.30 (0.42, 4.06) | 0.65 | 4.96 (0.52, 47.34) | 0.16 | ** | | 1.18 (0.71, 1.97) | 0.52 | 1.11 (0.67, 1.83) | 0.69 |
| **Delivery** | CRP | 0.83 (0.46, 1.50) | 0.54 | 1.02 (0.64, 1.63) | 0.93 | 1.19 (0.52, 2.74) | 0.68 | 0.26 (0.06, 1.02) | 0.055 | 0.70 (0.07, 6.66) | 0.78 | 0.95 (0.66, 1.37) | 0.78 | 0.85 (0.60, 1.20) | 0.35 |
| | AGP | 1.39 (0.82, 2.34) | 0.22 | 1.25 (0.84, 1.86) | 0.26 | **2.55 (1.27, 5.11)** | **0.0085** | 13.11 (1.62, 106.23) | 0.016 | ** | | **1.63 (1.20, 2.22)** | **0.0018** | 1.24 (0.90, 1.70) | 0.18 |
| **DP+AZ** | | | | | | | | | | | | | | | |
| **Enrolment (Week 16–28)** | CRP | 1.39 (0.82, 2.36) | 0.22 | 1.26 (0.87, 1.82) | 0.22 | 1.29 (0.69, 2.43) | 0.43 | 1.57 (0.40, 6.14) | 0.51 | ** | | 1.35 (0.99, 1.82) | 0.054 | 0.99 (0.73, 1.35) | 0.96 |
| | AGP | 1.37 (0.79, 2.38) | 0.26 | **1.49 (1.02, 2.17)** | **0.04** | 1.36 (0.70, 2.64) | 0.36 | 0.61 (0.12, 3.10) | 0.55 | ** | | **1.49 (1.09, 2.03)** | **0.012** | 0.89 (0.63, 1.24) | 0.49 |
| **Visit 3** | CRP | 0.97 (0.37, 2.53) | 0.95 | 1.20 (0.71, 2.02) | 0.50 | 0.74 (0.24, 2.32) | 0.60 | ** | | 5.03 (0.30, 83.44) | 0.26 | 1.22 (0.79, 1.88) | 0.37 | 0.97 (0.59, 1.57) | 0.89 |
| | AGP | **2.25 (1.02, 4.95)** | **0.044** | 1.65 (0.97, 2.80) | 0.06 | 0.43 (0.06, 3.04) | 0.40 | ** | | ** | | 1.41 (0.87, 2.29) | 0.17 | 1.04 (0.58, 1.86) | 0.89 |

*(Continued)*

**Table 4.** (Continued)

| SP | | LBW | | SGA | | PTB | | Foetal loss | | Neonatal loss | | Any adverse foetal outcome | | Low maternal weight gain | |
|---|---|---|---|---|---|---|---|---|---|---|---|---|---|---|---|
| | | aRR (95% CI) | P | aRR (95% CI) | P | aRR (95% CI) | P | aRR (95% CI) | P | aRR (95% CI) | P | aRR (95% CI) | P | aRR (95% CI) | P |
| Delivery | CRP | 0.87 (0.48 (1.60) | 0.66 | 1.09 (0.70, 1.69) | 0.71 | 0.57 (0.29, 1.12) | 0.10 | 0.41 (0.08, 2.28) | 0.31 | ** | | 0.94 (0.66, 1.33) | 0.72 | 1.00 (0.70, 1.45) | 0.98 |
| | AGP | 1.61 (0.93, 2.78) | 0.091 | 1.35 (0.92, 1.98) | 0.13 | **2.20 (1.10, 4.40)** | **0.027** | 0.96 (0.16, 5.65) | 0.97 | ** | | **1.57 (1.14, 2.15)** | **0.0057** | 1.24 (0.89, 1.72) | 0.20 |
| DP & DP+AZ | | | | | | | | | | | | | | | |
| Enrolment (Week 16–28) | CRP | 1.26 (0.87, 1.82) | 0.22 | 1.11 (0.85, 1.46) | 0.43 | 1.20 (0.77, 1.88) | 0.42 | 1.75 (0.68, 4.50) | 0.24 | ** | | 1.20 (0.97, 1.49) | 0.092 | 1.01 (0.81, 1.26) | 0.90 |
| | AGP | 1.40 (0.95, 2.06) | 0.09 | **1.35 (1.02, 1.79)** | **0.037** | 1.30 (0.81, 2.10) | 0.27 | 0.88 (0.30, 2.59) | 0.82 | ** | | **1.40 (1.12, 1.74)** | **0.033** | 1.02 (0.81, 1.29) | 0.88 |
| Visit 3 | CRP | **1.70 (1.04, 2.79)** | **0.035** | 1.31 (0.92, 1.87) | 0.13 | 1.00 (0.49, 2.05) | 0.99 | ** | | 2.78 (0.44, 17.39) | 0.28 | 1.33 (1.00, 1.77) | 0.053 | 1.03 (0.75, 1.42) | 0.84 |
| | AGP | **1.85 (1.04, 3.27)** | **0.036** | 1.37 (0.90, 2.08) | 0.14 | 0.81 (0.30, 2.16) | 0.68 | ** | | ** | | 1.30 (0.91, 1.84) | 0.16 | 1.08 (0.73, 1.58) | 0.70 |
| Delivery | CRP | 0.85 (0.56, 1.30) | 0.46 | 1.05 (0.76, 1.46) | 0.75 | 0.80 (0.48, 1.35) | 0.41 | **0.31 (0.10, 0.90)** | **0.03** | ** | | 0.94 (0.73, 1.22) | 0.66 | 0.92 (0.71, 1.18) | 0.51 |
| | AGP | **1.48 (1.01, 2.18)** | **0.044** | 1.30 (0.99, 1.71) | 0.064 | **2.37 (1.44, 3.90)** | **0.0007** | 3.83 (1.18, 12.38) | 0.025 | ** | | **1.60 (1.28, 2.00)** | **<0.0001** | 1.24 (0.99, 1.56) | 0.067 |

CRP, C-reactive protein; AGP, alpha-1-acid glycoprotein. SP, sulphadoxine-pyrimethamine; DP, dihydroartemisinin-piperaquine; AZ, azithromycin. Visit 3, third treatment with IPTp 24–36 gestation weeks). The composite adverse birth outcome variable includes: low birth weight (LBW) defined as <2500 g, preterm birth (PTB), defined as <37 gestational weeks, foetal loss defined as a non-live birth, small for gestational age (SGA) defined as birth weight lower than the 10[th] centile of the intergrowth-21[st] standards [1] and/or neonatal loss. Low gestational weight gain was defined as the bottom quartile of the IMPROVE study population (<164 g/week). Adjusted Risk Ratios (aRR) and 95% confidence intervals were estimated using logistic regression and adjusted for maternal age, maternal height, mid-upper arm circumference, gravidity, number of antenatal visits and study site, and included an interaction term between inflammatory marker and treatment arm. Significance is defined as P <0.05 and significant findings are highlighted with bold text. ** aRRs were not always able to be estimated for neonatal death or foetal loss as these outcomes occurred infrequently.

and gestation length in the DP arm (Table 5). In SP recipients, women with elevated AGP at enrolment also had increased risk of low maternal weight gain (aRR 1.94, 95% CI: 1.36, 2.76), and gained on average 0.090 Kg/week less than those without elevated AGP (95% CI: -140, -4). An association between elevated AGP at enrollment and weight gain was not seen in women receiving DP or DP+AZ (aRR 1.02, 95% CI: 0.81, 1.29) (Table 4).

At Visit 3, elevated CRP was weakly associated with an increased risk of LBW in women receiving SP and in those receiving DP (aRR 2.06, 95% CI: 0.97, 4.38 and aRR 2.44, 95% CI: 1.36, 4.37, respectively). Similar relationships were observed among those with elevated CRP for increased risks of SGA (SP aRR 1.80, 95% CI: 1.08, 2.97 and DP aRR 1.43, 95% CI: 0.90, 2.28). Elevated AGP at Visit 3 was associated wth increased risk of LBW in women receiving SP or DP+AZ (aRR 2.47, 95% CI: 1.05, 5.82 and aRR 2.25, 95% CI: 1.02, 4.95), and women receiving SP with elevated AGP at Visit 3 had babies born 6.19 days earlier than those without this inflammation (95% CI: -10.78, -1.61) (Table 4).

**Table 5. Elevated CRP and AGP levels during pregnancy and associations with clinical variables, by treatment arm.**

| SP | | Birth weight (kg) | | Gestational age days | | Growth z-score | | Maternal weight gain (kg/week) | |
|---|---|---|---|---|---|---|---|---|---|
| | | coefficient (95% CI) | P | coefficient (95% CI) | P | coefficient (95% CI) | P | coefficient (95% CI) | P |
| Enrolment (Week 16–28) | CRP | -0.07 (-0.17, 0.03) | 0.15 | 0.55 (-2.60, 3.70) | 0.73 | -0.17 (-0.39, 0.04) | 0.12 | 0.010 (-0.04, 0.06) | 0.68 |
| | AGP | **-0.17, (-0.27, -0.07)** | **0.001** | **-3.51, (-6.87, -0.15)** | **0.041** | -0.15 (-0.38, 0.08) | 0.21 | **-0.09 (-0.14, -0.04)** | **0.001** |
| Visit 3 | CRP | -0.07 (-0.19, 0.05) | 0.24 | -1.57 (-4.99, 1.86) | 0.37 | -0.10 (-0.37, 0.18) | 0.49 | -0.004 (-0.06, 0.06) | 0.90 |
| | AGP | -0.04 (-0.19, 0.12) | 0.63 | **-6.19 (-10.78, -1.61)** | **0.008** | 0.22 (-0.15, 0.58) | 0.25 | -0.04 (-0.12, 0.04) | 0.35 |
| Delivery | CRP | 0.03 (-0.08, 0.14) | 0.63 | -0.51 (-3.82, 2.81) | 0.76 | 0.18 (-0.05, 0.41) | 0.13 | -0.02 (-0.08, 0.03) | 0.45 |
| | AGP | 0.006 (-0.10, 0.11) | 0.90 | -0.17 (-3.31, 2.98) | 0.92 | 0.03 (-0.19, 0.24) | 0.81 | 0.002 (-0.05, 0.05) | 0.93 |
| **DP** | | | | | | | | | |
| Enrolment (Week 16–28) | CRP | 0.01 (-0.08, 0.10) | 0.84 | -0.83 (-3.88, 2.21) | 0.59 | 0.11 (-0.10, 0.32) | 0.31 | -0.03 (-0.07, 0.02) | 0.24 |
| | AGP | -0.08 (-0.18, 0.01) | 0.09 | 0.14 (-3.03, 3.31) | 0.93 | -0.20 (-0.42, 0.02) | 0.075 | -0.04 (-0.09, 0.01) | 0.088 |
| Visit 3 | CRP | -0.06 (-0.19, 0.07) | 0.40 | 0.35 (-3.34, 4.05) | 0.85 | -0.17 (0.47, 0.13) | 0.27 | 0.01 (-0.05, 0.08) | 0.64 |
| | AGP | -0.15 (0.31, 0.02) | 0.083 | -0.82 (-5.52, 3.89) | 0.73 | -0.28 (-0.67, 0.11) | 0.16 | 0.04 (-0.04, 0.12) | 0.33 |
| Delivery | CRP | 0.05 (-0.06, 0.16) | 0.39 | 0.84 (-2.51, 4.20) | 0.62 | 0.17 (-0.06, 0.41) | 0.15 | 0.04 (-0.02, 0.09) | 0.19 |
| | AGP | -0.07 (-0.17, 0.03) | 0.20 | -2.42 (-5.44, 0.61) | 0.12 | -0.03 (-0.24, 0.18) | 0.77 | -0.03 (-0.08, 0.02) | 0.31 |
| **DP+AZ** | | | | | | | | | |
| Enrolment (Week 16–28) | CRP | **-0.10 (-0.19, -0.002)** | **0.045** | -0.99 (-0.409, 2.11) | 0.53 | -0.18 (-0.39, 0.04) | 0.11 | -0.00 (-0.05, 0.04) | 0.89 |
| | AGP | **-0.13 (-0.23, -0.03)** | **0.01** | **-3.44 (-6.70, -0.17)** | **0.039** | -0.06 (-0.29, 0.17) | 0.60 | -0.02 (-0.07, 0.03) | 0.35 |
| Visit 3 | CRP | -0.06 (-0.18, 0.08) | 0.46 | 2.15 (-1.75, 6.04) | 0.30 | -0.23 (-0.55, 0.08) | 0.15 | **0.07 (0.00, 0.13)** | **0.048** |
| | AGP | -0.09 (-0.26, 0.08) | 0.29 | 1.16 (-3.68, 6.00) | 0.64 | -0.28 (-0.67, 0.11) | 0.16 | 0.02 (-0.06, 0.11) | 0.58 |
| Delivery | CRP | 0.05 (-0.06, 0.16) | 0.35 | **4.99 (1.73, 8.25)** | **0.003** | -0.11 (-0.33, 0.12) | 0.36 | 0.03 (-0.02, 0.08) | 0.28 |
| | AGP | 0.08 (-0.18, 0.02) | 0.12 | -1.45 (-4.51, 1.62) | 0.35 | -0.06 (-0.28, 0.15) | 0.56 | -0.04 (-0.09, 0.01) | 0.14 |
| **DP & DP+AZ** | | | | | | | | | |
| Enrolment (Week 16–28) | CRP | -0.04 (-0.11, 0.02) | 0.21 | -0.91 (-3.09, 1.27) | 0.41 | -0.03 (-0.18, 0.12) | 0.69 | -0.02 (-0.05, 0.02) | 0.36 |
| | AGP | **-0.11 (-0.18, -0.04)** | **0.003** | -1.59 (-3.93, 0.74) | 0.18 | -0.13 (-0.30, 0.03) | 0.11 | -0.03 (-0.07, 0.002) | 0.069 |
| Visit 3 | CRP | -0.05 (-0.15, 0.04) | 0.26 | 1.22 (-0.50, 3.93) | 0.38 | -2.01 (-0.42, 0.02) | 0.075 | 0.04 (-0.01, 0.09) | 0.10 |
| | AGP | **-0.12 (-0.24, -0.001)** | **0.049** | 0.15 (-3.25, 3.54) | 0.93 | **-0.28 (-0.56, -0.002)** | **0.048** | 0.03 (-0.03, 0.09) | 0.29 |
| Delivery | CRP | 0.05 (-0.3, 0.13) | 0.21 | **2.96 (0.61, 5.31)** | **0.014** | 0.03 (-0.14, 0.19) | 0.78 | 0.03 (-0.005, 0.07) | 0.092 |
| | AGP | **-0.07 (-0.14, -0.002)** | **0.043** | -1.93 (-4.10, 0.23) | 0.080 | -0.05 (-0.20, 0.10) | 0.54 | -0.03 (-0.07, 0.004) | 0.082 |

CRP, C-reactive protein; AGP, alpha-1-acid glycoprotein. SP, sulphadoxine-pyrimethamine; DP, dihydroartemisinin-piperaquine; AZ, azithromycin. Visit 3, third treatment with IPTp 24–36 gestation weeks). Coefficient and 95% CI were calculated using linear regression adjusted for maternal age, heigh, mid-upper arm circumference, gravidity, number of antenatal visits and study site, with an interaction term included between inflammatory marker and treatment arm. Significance is defined as P < 0.05 and significant findings are highlighted with bold text.

At delivery, elevated AGP was not associated with adverse birth outcomes in women receiving SP (Table 4). By contrast, in women receiving either DP or DP+AZ, elevated AGP at delivery was associated with increased risks of LBW (aRR 1.48, 95% CI: 1.01, 2.18), PTB (aRR 2.37, 95% CI: 1.44, 3.90), foetal loss (aRR 3.83, 95% CI: 1.18, 12.38) and evidence of an increased risk for SGA (aRR 1.30, CI 0.99, 1.71), however within the individual DP and DP+AZ groups elevated AGP at delivery was not significantly associated with foetal loss or low birthweight. Women receiving either DP or DP+AZ with elevated AGP had babies 70 g smaller than those without elevated AGP (-95% CI: -140, -0.2) (Table 5) and elevated CRP at delivery was associated with a decreased risk of foetal loss in women receiving either DP or DP+AZ (aRR 0.31, 95% CI: 0.10, 0.90) (Table 4) (note neither of these associations were significant in the individual DP or DP+AZ groups). Elevated CRP at delivery was also associated with a longer gestation in women receiving DP+AZ (4.99 days, 95% CI: 1.73, 8.25) (Table 5).

**Table 6. Ratios of risk ratios describing the association between elevated inflammatory markers and adverse birth outcomes by treatment arms.**

| | | DP VS SP | P | DP+AZ VS SP | P | DP VS DP+AZ | P | DP & DP+AZ VS SP | P |
|---|---|---|---|---|---|---|---|---|---|
| Enrolment (Week 16–28) | CRP | 0.88 (0.56, 1.37) | 0.56 | 1.09 (0.70, 1.71) | 0.70 | 0.80 (0.52, 1.20) | 0.31 | 0.98 (0.66, 1.45) | 0.90 |
| | AGP | 0.94 (0.60, 1.46) | 0.78 | 1.06 (0.68, 1.67) | 0.79 | 0.88 (0.58, 1.34) | 0.56 | 1.00 (0.67, 1.48) | 0.98 |
| Visit 3 | CRP | 0.96 (0.56, 1.66) | 0.88 | 0.82 (0.45, 1.47) | 0.50 | 1.18 (0.67, 2.07) | 0.57 | 0.89 (0.55, 1.46) | 0.65 |
| | AGP | 0.80 (0.39, 1.63) | 0.54 | 0.95 (0.48, 1.89) | 0.89 | 0.84 (0.42, 1.69) | 0.63 | 0.87 (0.47, 1.60) | 0.66 |
| Delivered | CRP | 1.11 (0.65, 1.91) | 0.70 | 1.10 (0.64, 1.87) | 0.73 | 1.01 (0.61, 1.68) | 0.96 | 1.11 (0.69, 1.78) | 0.67 |
| | AGP | **1.99 (1.18, 3.34)** | **0.01** | **1.91 (1.13, 3.23)** | **0.016** | 1.04 (0.67, 1.62) | 0.86 | **1.95 (1.21, 3.13)** | **0.006** |

CRP, C-reactive protein; AGP, alpha-1-acid glycoprotein. SP, sulphadoxine-pyrimethamine; DP, dihydroartemisinin-piperaquine; AZ, azithromycin. The measures of the association in each cell represent the ratio (95% CI) of the adjusted risk ratio of the association between elevated CRP/AGP levels and the composite adverse pregnancy outcome obtained from logistic regression models with an interaction term between treatment arm and CRP/AGP, adjusted for maternal age, maternal height, mid-upper arm circumference, gravidity, number of antenatal visits and study site. Significance is defined as P < 0.05 and significant findings are highlighted with bold text.

When we compared the risk of an association between any adverse birth outcome and inflammation in the different treatment arms there was little evidence of differences in risk of an association between elevations of CRP or AGP and adverse birth outcomes in the SP, DP or DP+AZ arms at enrollment or visit 3 (all P>0.3) (Table 6). The risk of associations between elevated CRP at delivery and outcome were also similar between the three arms. However at delivery, associations between elevated AGP and outcome were almost twice as likely to be observed in the DP or DP+AZ arms compared to the SP arm (aRR 1.95, 95% CI: 1.21, 3.13) (Table 6).

## Discussion

This study investigated whether SP modified the associations between elevated biomarkers AGP and CRP and adverse birth outcomes. Elevated AGP concentrations were more commonly associated with adverse birth outcomes than elevated CRP. Associations between inflammation and adverse outcomes were seen in all three arms at enrolment, while at delivery, these associations were restricted to the DP+AZ and DP arms (note, some associations reached significance only in the combined DP & DP+AZ group), suggesting that SP may dampen the relationship between inflammation at delivery and adverse birth outcomes.

There were clear associations between inflammatory biomarker AGP and birth outcomes in this cohort. CRP and other inflammatory biomarkers such as pro-inflammatory cytokines, together with growth factors and hormones, have been associated with SGA and PTB in a meta-analysis that largely comprised studies from high-income countries [23]. Interestingly, AGP did not feature in this meta-analysis. In malaria-endemic Malawi, there were few associations between CRP elevation and adverse birth outcomes over the three time points, consistent with a study from PNG [17] and a study in HIV-infected pregnant women from Tanzania [18], in both of which CRP concentrations were not associated with LBW, SGA or PTD. The lack of associations with elevated CRP might reflect the timing of sampling (which was later in pregnancy in the LMIC studies) or may be driven by the higher prevalence of infectious diseases in LMICs, and the lower prevalence of obesity, which can also elevate CRP [24].

Elevated AGP at enrolment was associated with the composite adverse pregnancy outcome, SGA and low gestational weight gain. Similarly, in PNG, higher concentrations of AGP at delivery were associated with LBW, PTB and SGA, and in Tanzania increased AGP at 32 weeks' gestation was associated with PTB and lower birth weight [18]. In Nepal, AGP (but not CRP) concentration in first or third trimester was inversely associated with birth weight,

length, head circumference (third trimester only) and chest circumference [25]. Also in Nepal, AGP concentrations at 32 weeks were higher in women who gave birth to LBW or PTB babies compared to those with normal birth weight term deliveries [26]. Elevated AGP may be a better marker than elevated CRP of persistent inflammation, which may contribute more to adverse birth outcomes than intense inflammatory responses of shorter duration [27]. Together, these findings suggest that AGP should be studied further as a marker of adverse pregnancy outcomes in LMICs, and that understanding the drivers of increased AGP concentration may reveal important pathways for intervention to minimize adverse pregnancy outcomes.

Associations between inflammatory markers and maternal weight gain varied with IPTp regime. In women receiving SP, poor maternal weight gain was associated with elevated AGP at enrolment. Overall maternal weight gain was highest in women receiving SP [19], and in PNG IPTp with SP and AZ was associated with greater maternal weight gain than single dose SP and chloroquine [17]. These findings suggest that some women respond to SP with increased weight gain.

Concentrations of CRP and AGP fell between enrolment and visit 3, when there were few associations between inflammation and adverse outcomes and comparatively few women had inflammation, and then rose again at delivery. The mechanism underlying these changes are unclear, but similar declines in AGP between the end of the first and second trimester were reported from a study of pregnant women in Pakistan, India and Guatemala who did not receive IPTp [28] and an increase in CRP in the third trimester (compared to the first) has been previously seen in a pregnant women from Nepal [25].

At delivery, elevated AGP was associated with adverse outcomes in DP and DP+AZ recipients and not in women receiving SP, with the risk of an association between inflammation and adverse outcome twice as high in the DP or DP+AZ arms compared to the SP arm. This suggests that SP modifies or resolves a risk factor for adverse outcomes, which remains operative in women receiving DP or DP+AZ. It is unlikely that the effect is due to SP simply dampening the immune response as concentrations of AGP declined more over follow up in the DP+AZ arm than in the SP arm, an observation that is consistent with the known immunomodulatory activity of AZ [29] and the number of women with elevated AGP at delivery was similar between the three arms. It is not related to clearance of *Plasmodium spp.* infections as the IPTp regimes with DP and DP+AZ had more potent anti-malarial activity compared to SP [19]. Recent evidence using in-vitro models suggests that SP may directly improve the ability of the intestine to absorb nutrients in a nutrient deficient environment and suppresses the inflammatory responses associated with nutrient deficiency [30]. Also, unlike DP, which is specifically an anti-malarial, SP is an anti-folate drug with broad-spectrum antimicrobial properties, and may be clearing pathogens including STIs, such as the parasite *Chlamydia trachomatis* [31]. SP might also alter the gut microbiome, and so increase maternal weight gain [32].

Imbalances in maternal gut, oral and vaginal microbiomes have also been associated with PTB and preeclampsia, which is associated with chronic immune activation within the mother [33]. IPTp-SP has been associated with decreased incidence of new Enteroaggregative *E. Coli* (EAEC) infections, increased maternal weight gain and improved birth weight [32], which was restricted to women without these gut microbes at study entry. These findings would be consistent with a similar effect mediated through gut flora and nutrient absorption. SP might be affecting the gut microbiome in other ways and modifying inflammatory processes through pathways such as cytokine responses. However, the influence of the maternal intestinal microbiome on birth outcomes has yet to be fully classified [34]. The presence of certain vaginal organisms is predictive of PTB [35], and the vaginal microbiome could also be modified by SP to prevent adverse birth outcomes [36]. SP may be clearing specific pathogens including STIs

such as *Chlamydia trachomatis* which is associated with adverse birth outcomes [37]. It should, however, be borne in mind that AZ also has broad-spectrum antibiotic activity against both bacterial enteric pathogens and some prevalent STIs [38].

This study raises the idea that different IPTp regimes differentially modulate factors causing inflammation in pregnancy and may have differential effects on birth outcomes, although the mechanisms by which these effects operate require further study. This study's strengths include prospectively examining the relationship between inflammation and adverse outcomes at multiple time points in a cohort of women receiving IPTp with SP, DP or DP+AZ. Study limitations include the restriction to just two biomarkers, and collection of samples at delivery (when parturition could induce changes in the biomarkers), rather than late in gestation. These parturition mediated effects could result in misclassification of participants. Future studies of the associations between inflammatory markers and outcomes should include samples obtained late in gestation before parturition, and examine a broader range of biomarkers. Ongoing studies are examining the carriage of enteric pathogens, the maternal gut and vaginal microbiomes and STIs to further explore how associations between inflammation differed by treatment arm. After stratifying by treatment arm, the numbers of women with specific adverse outcomes such as foetal loss were small, requiring us to combine the outcome variables for the primary stratified analysis.

These results support the use of AGP as a prognostic inflammatory biomarker in pregnancy and suggest that SP receipt modulates the association between AGP as a marker of inflammation and outcomes. Further studies using a wider range of inflammatory markers would be of interest. The drivers of inflammation differ between the SP and non-SP arms and their concentrations vary over gestation. Future studies should address the importance of inflammation in adverse pregnancy outcomes and elucidate the pathways through which SP might modulate this relationship to improve pregnancy outcomes, incorporating assessment of malaria, STIs and the maternal gut and vaginal microbiomes.

## Supporting information

**S1 Data. AGP and CRP levels at enrollment, visit 3 and delivery.**
(XLS)

**S1 Text. Global inclusivity questionnaire.**
(DOCX)

## Acknowledgments

We would like to thank all the women who participated in the IMPROVE study and donated their plasma samples for this study, and all the physicians and nurses involved in the study.

## Author Contributions

**Conceptualization:** Stephen J. Rogerson.

**Data curation:** James Dodd.

**Formal analysis:** Elizabeth H. Aitken.

**Funding acquisition:** Feiko O. ter Kuile, Stephen J. Rogerson.

**Investigation:** Kaylene Cheng, Wina Hasang.

**Methodology:** Kaylene Cheng, Wina Hasang, Niamh Meagher, David J. Price, Feiko O. ter Kuile.

**Resources:** Mwayiwawo Madanitsa, Victor Mwapasa, Kamija S. Phiri, Feiko O. ter Kuile.

**Supervision:** Elizabeth H. Aitken, Stephen J. Rogerson.

**Writing – original draft:** Kaylene Cheng, Elizabeth H. Aitken, Stephen J. Rogerson.

**Writing – review & editing:** Kaylene Cheng, Elizabeth H. Aitken, Feiko O. ter Kuile, Stephen J. Rogerson.

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
