## [Decision Letter · Decision Letter 0]

7 Feb 2024

PGPH-D-23-02194

Intermittent preventive treatment with sulphadoxine-pyrimethamine but not dihydroartemisinin-piperaquine modulates the relationship between inflammatory markers and adverse pregnancy outcomes in Malawi

Dear Dr. Aitken,

Thank you for submitting your manuscript to PLOS Global Public Health. After careful consideration, we feel that it has merit but does not fully meet PLOS Global Public Health’s publication criteria as it currently stands. Therefore, we invite you to submit a revised version of the manuscript that addresses the points raised during the review process.

We look forward to receiving your revised manuscript.

Kind regards,

Alassane Dicko

Academic Editor

Journal Requirements:

2. Please include a complete copy of PLOS’ questionnaire on inclusivity in global research in your revised manuscript. Our policy for research in this area aims to improve transparency in the reporting of research performed outside of researchers’ own country or community. The policy applies to researchers who have travelled to a different country to conduct research, research with Indigenous populations or their lands, and research on cultural artefacts. The questionnaire can also be requested at the journal’s discretion for any other submissions, even if these conditions are not met.  Please find more information on the policy and a link to download a blank copy of the questionnaire here: https://journals.plos.org/globalpublichealth/s/best-practices-in-research-reporting. Please upload a completed version of your questionnaire as Supporting Information when you resubmit your manuscript.

3. Please provide separate figure files in .tif or .eps format only and remove any figures embedded in your manuscript file. Please also ensure all files are under our size limit of 10MB.

4. We do not publish any copyright or trademark symbols that usually accompany proprietary names, eg  ©, ®, ™  (e.g. next to drug or reagent names). Please remove all instances of trademark/copyright symbols throughout the text, including ® & ™ on page 6.

5. In the online submission form, you indicated that "The data that support the findings of this study are available from the corresponding author upon reasonable request". All PLOS journals now require all data underlying the findings described in their manuscript to be freely available to other researchers, either 1. In a public repository, 2. Within the manuscript itself, or 3. Uploaded as supplementary information.

**Comments to the Author**

Reviewer #1: In this manuscript, the authors aimed to investigate if SP alters the relationship between inflammation and birth outcomes. The study describes that elevated AGP levels at delivery were associated with adverse birth outcomes in women that received DP or DP+AZ and concluded that SP modifies the relationship between inflammation and adverse outcomes.

One of my major comments is that the proportions of women with elevated AGP and CRP were similar between groups (Table 2), therefore, how does SP modify the relationships between inflammatory markers and outcomes if proportion of women with elevated levels are similar? For SP to change/reduce the risks, wouldn’t that be associated or even required to have lower levels of inflammatory markers? The authors should also show if CRP and AGP levels (as continuous variables) are similar between groups at the 3 time points.

Comments

1. Abstract, lines 51-52: without looking at the tables, this sentence implies that there are no differences between the groups. However, according to Table 4, elevated AGP levels at enrollment in the DP+AZ group was associated with increased SGA, and in the DP group, elevated AGP did not increased the risk of any adverse outcome.

2. Table 1: Are there any significant differences in birth outcomes (proportions) between the groups?

3. Infection status: what was the infection rate at the 3rd visit? At delivery, do the authors have placental blood smear or PCR results that can be included in the analysis.

4. How do you explain the reduction in AGP and CRP at visit 3 that was close to delivery in some, followed by increased levels at delivery?

5. Table 4, enrollment: because samples at enrollment were collected prior to drug administration, why AGP levels are associated with SGA or any adverse effect only in the DP+AZ group? How do you explain differences at enrollment based on treatment groups?

6. Similarly in Table 5, AGP levels at enrollment were associated with reduction in birthweight and gestational age in SP and DP+AZ groups but not in DP, and lower maternal weight gain in the SP group only, why is that? CRP levels at enrollment were associated with a reduction in birthweight only in the DP+AZ group but not in other treatment groups, please explain.

7. There is no reference in the paper to the increased gestational age associated with CRP at visit 3 in the DP+AZ group. The authors should comment on that.

8. Lines 288-294: elevated AGP levels at delivery were associated with increased LBW, PTB, foetal loss, reduced birthweight and gestational age after combining the DP with the DP+AZ group. This is not clear in the text. Please clarify that these associations were not observed in each of the groups alone.

9. Lines 347-348: “Associations between inflammation and adverse outcomes were seen in all three arms at enrolment”, this isn’t the case in the DP group (Tables 4 and 5)

10. Line 348-349: for clarity, the authors should change to “restricted to the DP+AZ and DP arms combined”. Similarly, clarify in the Abstract (line 54) that the results are for the 2 groups combined.

11. Lines 361-367: in the current study AGP at enrollment was associated with a composite of adverse outcomes, and in PNG at delivery. Is it correct that in the current study, AGP levels at delivery (all women) were not associated with adverse outcomes? At what time point AGP levels measured in the Tanzanian study?

12. Lines 412-414: Previous studies described increased levels of acute phase proteins at delivery in women that had spontaneous vaginal delivery vs C-section, the models can be adjusted for that if applicable here.

13. Did the authors examined the effect of malaria infection on the association between inflammatory markers and birth outcomes, i.e., including malaria infection as a covariate in the models.

Minor comments

1. Line 97: “a subset of IMPROVE cohort”, please describe how the subset was selected.

2. A new article presenting a model study to explain the mechanism associated with increased birthweight in women that received IPTp-SP was recently published (https://doi.org/10. 1016/j.ebiom.2023. 104921). The authors should consider referencing the study.

Reviewer #2: The authors address the modulation of the inflammation by the antimalarial compounds given as intermittent preventive treatment. This is a very important topic as there is strong evidence that SP-IPTp has significantly contributed to the reduction of low birth weight and other adverse pregnancy outcomes. With the occurrence of resistance to SP, alternatives are being evaluated, but recent studies have still shown that although SP did not have a strong antimalarial effect as mefloquine or dihydroartemisinin-piperaquine, it still showed good effect in preventing low birth weight, suggesting there could be another mechanism beyond the prevention of malaria.

he authors have assessed the CRP and AGP as inflammatory markers and they have well analysed and described theur findings and also pointed out the limitations of their analysis and suggested further studies to continue to explore the question. The findings are worth being published.

The following minor comments need to be addressed:

Be consistent with the date format. See line 110 and line 117.

Line 209: correct anti-natal

Line210: correct Kenua

From line 220: consider providing also absolute values within brackets so that the reader can check. For example instead of 35.7%... only, consider 35.7% (xx/xx)...

Reviewer #3: This manuscript reports result of a study on “Intermittent preventive treatment with sulphadoxine-pyrimethamine but not dihydroartemisinin-piperaquine modulates the relationship between inflammatory markers and adverse pregnancy outcomes in Malawi”, by Kaylene et al.

They found that Elevated alpha-1-acid glycoprotein (AGP)AGP at enrollment was associated with adverse birth outcome, with similar associations observed across treatment arms, and

with low maternal weight gain in SP recipients only. At visit 3 there were few associations between inflammation and adverse outcomes. At delivery, women with elevated AGP receiving DP or DP+AZ had an increased risk of adverse birth outcomes, including LBW, PTB and foetal loss. The risk of an association between elevated AGP and adverse birth outcome was higher in those receiving DP or DP+AZ compared to those receiving SP. In contrast to AGP, no clear associations between CRP and adverse outcomes were observed. The authors concluded that SP modifies the relationship between inflammatory biomarkers and adverse outcomes.

Summary:

This manuscript describes the results of a study on “Intermittent preventive treatment with sulphadoxine-pyrimethamine but not dihydroartemisinin-piperaquine modulates the relationship between inflammatory markers and adverse pregnancy outcomes in Malawi”, by Kaylene et al.

They assessed the Relationship between inflammation and birth outcomes in women receiving IPTp using Sulfadoxine-pyrimethamine or dihydroartemisinin piperaquine.

The results indicate there that Elevated alpha-1-acid glycoprotein (AGP)AGP at enrollment was associated with adverse birth outcome, with similar associations observed across treatment arms, and

with low maternal weight gain in SP recipients only. At visit 3 there were few associations between inflammation and adverse outcomes. At delivery, women with elevated AGP receiving DP or DP+AZ had an increased risk of adverse birth outcomes, including LBW, PTB and foetal loss. The risk of an association between elevated AGP and adverse birth outcome was higher in those receiving DP or DP+AZ compared to those receiving SP. In contrast to AGP, no clear associations between CRP and adverse outcomes were observed. The authors concluded that SP modifies the relationship between inflammatory biomarkers and adverse outcomes.

General comments:

The results indicate there that Elevated alpha-1-acid glycoprotein (AGP)AGP at enrollment was associated with adverse birth outcome, with similar associations observed across treatment arms, and

with low maternal weight gain in SP recipients only. At visit 3 there were few associations between inflammation and adverse outcomes. At delivery, women with elevated AGP receiving DP or DP+AZ had an increased risk of adverse birth outcomes, including LBW, PTB and foetal loss. The risk of an association between elevated AGP and adverse birth outcome was higher in those receiving DP or DP+AZ compared to those receiving SP. In contrast to AGP, no clear associations between CRP and adverse outcomes were observed. The authors concluded that SP modifies the relationship between inflammatory biomarkers and adverse outcomes and suggested that their findings provide insights into the potential mechanisms by which SP may improve pregnancy outcomes.

Specific comments:

1) Abstract: Can the authors spelled out the terms LBW, PTB?

2) Introduction:

Line 69: birthweigh is in single word, but in two words in other places (eg line 68). Please be consistent.

Line 75: Can the authors add a reference at the end of the sentence?

3) Methods:

Line 119: can the authors add the reference of the approval letter for COMREC?

Line 123. Sex is listed as variable that was recorded at birth. However, this variable is not presented anywhere in the results section. Is that variable less important to modify your conclusion?

Line 132: A reference is needed at the end of this sentence.

Line 133: The authors modified the protocols for use with 384 microwell plates. Can the authors explain the rational of that modification?

Line 162: The authors mentioned that Changes in inflammatory markers in 162 individual women were calculated by subtracting levels at enrollment from those at visit 3. Did the authors also assess the same changes from enrollment to delivery? If not, one would like to know that change.

4. Results:

Table 1:

What is the rational of measuring the Mid upper arm circumference? This measure is not seen in any of the advance analysis. Same for the small for gestational age under the birth outcomes.

Notes underneath of table: Low birth weight (line 210) is in two words, but in one word in other places in the text. The

---

## [Decision Letter · Decision Letter 1]

12 Apr 2024

Intermittent preventive treatment with sulphadoxine-pyrimethamine but not dihydroartemisinin-piperaquine modulates the relationship between inflammatory markers and adverse pregnancy outcomes in Malawi

PGPH-D-23-02194R1

Dear Dr Aitken,

We are pleased to inform you that your manuscript 'Intermittent preventive treatment with sulphadoxine-pyrimethamine but not dihydroartemisinin-piperaquine modulates the relationship between inflammatory markers and adverse pregnancy outcomes in Malawi' has been provisionally accepted for publication in PLOS Global Public Health.

Best regards,

Alassane Dicko

Academic Editor

Reviewer Comments (if any, and for reference):

Reviewer's Responses to Questions

**Comments to the Author**

1. If the authors have adequately addressed your comments raised in a previous round of review and you feel that this manuscript is now acceptable for publication, you may indicate that here to bypass the “Comments to the Author” section, enter your conflict of interest statement in the “Confidential to Editor” section, and submit your "Accept" recommendation.

Reviewer #1: All comments have been addressed

2. Does this manuscript meet PLOS Global Public Health’s publication criteria? Is the manuscript technically sound, and do the data support the conclusions? The manuscript must describe methodologically and ethically rigorous research with conclusions that are appropriately drawn based on the data presented.

Reviewer #1: Yes

3. Has the statistical analysis been performed appropriately and rigorously?

Reviewer #1: Yes

4. Have the authors made all data underlying the findings in their manuscript fully available (please refer to the Data Availability Statement at the start of the manuscript PDF file)?

Reviewer #1: Yes

5. Is the manuscript presented in an intelligible fashion and written in standard English?

Reviewer #1: Yes

6. Review Comments to the Author

Reviewer #1: The authors provided answers to the comments

7. PLOS authors have the option to publish the peer review history of their article (what does this mean?). If published, this will include your full peer review and any attached files.

**Do you want your identity to be public for this peer review?** For information about this choice, including consent withdrawal, please see our Privacy Policy.

Reviewer #1: No
